# Real-World Analysis on the Characteristics, Therapeutic Paths and Economic Burden for Patients Treated for Glaucoma in Italy

**DOI:** 10.3390/healthcare11050635

**Published:** 2023-02-21

**Authors:** Valentina Perrone, Dario Formica, Benedetta Piergentili, Luca Rossetti, Luca Degli Esposti

**Affiliations:** 1CliCon s.r.l. Società Benefit-Health, Economics & Outcomes Research, 40137 Bologna, Italy; 2AbbVie S.r.l., SR 148 Pontina, 04011 Campoverde, Italy; 3Department of Ophthalmology, University of Milan San Paolo Hospital, 20142 Milan, Italy

**Keywords:** glaucoma, antiglaucoma preparations, real-world analysis

## Abstract

This real-world analysis was performed on administrative databases to evaluate characteristics, therapies, and related economic burden of glaucoma in Italy. Adults with at least 1 prescription for ophthalmic drops (ATC class S01E: antiglaucoma preparations, miotics) during data availability period (January 2010−June 2021) were screened, then patients with glaucoma were included. First date of ophthalmic drops prescription was the index date. Included patients had at least 12 months of data availability before index-date and afterwards. Overall, 18,161 glaucoma-treated patients were identified. The most frequent comorbidities were hypertension (60.2%), dyslipidemia (29.7%) and diabetes (17%). During available period, 70% (*N* = 12,754) had a second-line therapy and 57% (*N* = 10,394) a third-line therapy, predominantly ophthalmic drugs. As first-line, besides 96.3% patients with ophthalmic drops, a small proportion reported trabeculectomy (3.5%) or trabeculoplasty (0.4%). Adherence to ophthalmic drops was found in 58.3% patients and therapy persistence reached 78.1%. Mean total annual cost per patient was €1,725, mostly due to all-cause drug expenditure (€800), all-cause hospitalizations (€567) and outpatient services (€359). In conclusion, glaucoma-treated patients were mostly in monotherapy ophthalmic medications, with an unsatisfying adherence and persistence (<80%). Drug expenditures were the weightiest item among healthcare costs. These real-life data suggest that further efforts are needed to optimize glaucoma management.

## 1. Introduction

Vision impairment represents an important public health issue, and its burden is likely to increase in the future because of ageing of the global population [1]. Glaucoma is a chronic optic neuropathy age-related and among the main causes of vision loss [2]. The characteristic progressive damage of the optic nerve leads to an irreversible, although preventable, visual field loss [3]. Generally, the symptoms are almost absent at early stages and arise at late stages with problems related to permanent visual loss [4]. To date, the only controllable factor to prevent or delay the progressive course of glaucoma is the elevated intraocular pressure (IOP), even though studies suggest other modifiable risk factor could be represented by socioeconomic status, dietary intake, poor exercise or sleeping apnea [5].

Last estimates indicate approximately 60 million individuals with glaucoma worldwide, and around 8 million for Europe, with a prevalence of 2.5% [6]. In Italy around 550,000 individuals are estimated to have received a diagnosis for glaucoma [7]. The potential blindness, as well as the irreversible vision impairment, have a detrimental impact on the quality of life of glaucoma patients, that has been reported to decrease in parallel with the increment of glaucoma severity [8,9].

Glaucoma is often underdiagnosed, or diagnosis occurs at a later stage [10]. Antiglaucoma treatments aim to reduce and prevent further damage to the optic nerve and to preserve the residual visual capacity [7]. The most recent European Glaucoma Society (ESG) Guidelines advise the strategy proven to be effective focuses on lowering IOP. Treatments available are represented by medication, laser or surgery [7,10]. Pharmacological treatments, i.e., topical ophthalmic drops as monotherapy are considered to be first line therapy [11]. In case of lack of efficacy or intolerance, switching to a second drug in monotherapy or combination is advised. Among second line option, trabeculoplasty may also be considered, and the most recent guidelines recommended that trabeculoplasty should be considered as an option for initial treatment in mild or moderate phases of open angle glaucoma [11]. For patients using medication, ensuring an optimal level of adherence and persistence to treatments is essential to reduce risk of disease progression [12,13]. Indeed, good adherence and persistence are key points to obtain the beneficial effect of glaucoma therapy, by lowering IOP to prevent vision loss. Sub-optimal levels of adherence and persistence represent risk factors for disease progression, and it has been described in the literature that patients with stable visual fields were more than 75% adherent to their therapy, while patients with a worsening of their condition were less than 45% adherent [14]. Large evidence showed poor adherence to the prescribed topical drops for glaucoma treatment, when compared to medication adherence for other systemic chronic conditions [13,15,16]. Several methodologies for adherence evaluation have been reported, some of which are self-report, pharmacy refill reports, electronic monitoring and direct observation [15], but to date there is not a clear pattern on what method correlates best with clinically outcomes. Moreover, instrument scoring systems have been introduced and have been shown to predict the actual glaucoma medication adherence [17].

To date, little evidence is available on the drug utilization, characteristics and economic burden of patients with glaucoma in Italy. Hence, the analysis aims to evaluate the characteristics of patients with glaucoma, to describe their diagnostic and therapeutic paths, to assess the drug utilization of ophthalmic drops used by these patients and to analyze the health care resource use and related direct costs for Italian National Health Service (INHS) in clinical practice in Italy.

## 2. Materials and Methods

### 2.1. Data Source

This is a retrospective observational analysis that used data collected from the administrative health databases of Italian Local Health Units (LHUs) from Puglia, Campania, Umbria, Lazio and Veneto Regions, covering around 2.7 million health-assisted subjects. Such databases store information on all healthcare resources reimbursed by the INHS. The databases used to perform the analysis were: demographic database (to get data on age and sex), pharmaceutical database (with data related to drugs dispensed, such as Anatomical–Therapeutic Chemical [ATC] code, number of packages, number of units per package, costs and prescription date), hospitalization database (including discharge diagnosis codes classified according to the International Classification of Diseases, Ninth Revision, Clinical Modification [ICD-9-CM], diagnosis Related Group [DRG] and DRG related charge), outpatient specialist services database (containing data on type, description activity of diagnostic tests and specialist visits for patients in analysis) and payment exemption database (containing date and type of exemption).

An anonymous univocal patient ID was assigned by the LHUs to each health-assisted subject to ensure patient privacy. This ID allowed us to perform the electronic linkage between the databases. The anonymity process was in full compliance with UE Data Privacy Regulation 2016/679 (“GDPR”) and Italian D.lgs. n. 196/2003, as amended by D.lgs. n. 101/2018. Aggregated results are herein reported, so that it is not possible to connect to individual patients. 

### 2.2. Patient Population

All records of adult patients (≥18 years old) with at least 1 prescription for ophthalmic drops belonging to the class of ATC S01E (antiglaucoma preparations and miotics) during all data availability period, which spanned from January 2010 to June 2021, were screened for inclusion. Among them, patients with glaucoma were detected during the inclusion period January 2011−June 2020 by the presence of at least one of the following criteria (not necessarily after the ophthalmic drops prescription): (i) presence of hospitalization discharge diagnosis for glaucoma (ICD-9-CM: 365); (ii) an active exemption code for glaucoma (code 019); (iii) procedure for trabeculectomy (codes 12.64 OR 12.54) or trabeculoplasty (code 12.59) (as proxy of diagnosis). The index date was the date of the first ophthalmic drops prescription. All patients included in the analysis had at least 12 months of data availability period prior and afterward the index date, while those with missing data were excluded. Follow-up went from index date to end of data availability period or death (whichever occurred first).

### 2.3. Baseline Patient’ Characteristics

At index date, data on age and sex were analyzed. Presence of comorbidities was investigated in the year prior index date by evaluating the Charlson Comorbidity index [18], which gives a score based on the presence of specific comorbidities identified by hospitalization discharge diagnosis and/or drugs treatment (therefore, untreated/hospitalized comorbidities are not captured). Moreover, the proportion of patients affected by the following conditions has been reported: hypertension (at least one antihypertensive drugs prescription, ATC codes: C02, C03; C07; C08; C09), dyslipidemia (at least one lipid modifying agents prescription, ATC code: C10); diabetes (at least one antidiabetic drugs prescription, ATC code A10); cataract (ICD-9-CM code 366 or procedure codes 13.2, 13.3, 13.4, 13.6, 13.71); blindness (ICD-9-CM code 369 or exemption code C05); retinal/choroid disorders (ICD-9-CM code 361, 362, 363); diabetic retinopathy (DR): (ICD-9-CM code: 362.0); wet age-related macular degeneration (wAMD) (ICD-9-CM code 362.52); retinal vein occlusion (RVO) (ICD-9-CM code 362.3), Parkinson’s disease (ICD-9-CM code 332 or exemption code 038); Alzheimer’s disease (ICD-9-CM code 331.0 or exemption code 029); rheumatoid arthritis (ICD-9-CM code 714.0 or exemption code 006). Since comorbidities were identified based on hospitalization/treatment reimbursed by the INHS, they could have been underestimated.

Follow-up. Treatment line identification was performed and considered the whole analysis period. The number of lines was identified by presence of ophthalmic drops alone or in combination. Switching from one ophthalmic agent to another one was defined as change of line. Trabeculectomy and trabeculoplasty were considered as distinct treatment lines. The drug utilization was assessed by evaluating persistence, adherence and discontinuation of ophthalmic drops. Specifically, persistence was defined as presence of any ophthalmic drop prescriptions during the last quarter of 12 months follow-up. Discontinuation was identified as the absence of ophthalmic drops treatment prescriptions during the last trimester of 12 months follow-up period (interruption) or switching to another ophthalmic drops treatment (switch). Adherence to ophthalmic drops treatment was calculated during the first 12 months of follow-up by using the proportion of days covered (PDC), i.e., the ratio between the number of days of medication supplied and the observed time. Patients were classified as adherent (PDC ≥ 80), partially adherent (40 ≤ PDC < 80) and poorly adherent (PDC < 40%) [13]. Adherence was calculated based on prescriptions, and the actual use made by the patient is unknown.

### 2.4. Healthcare Resource Consumption and Costs

The analyses on healthcare resource consumption and costs were performed over the first year of follow-up on alive patients. Healthcare resource consumptions were reported as annual mean (and standard deviation, SD) number of all drug prescriptions, all-cause hospitalizations, all outpatient services per patient. Direct medical costs related to the healthcare resource consumption described above were reported in Euros (€) as annual mean with SD cost per patient. Drug costs were evaluated based on the INHS purchase price. Hospitalization costs were determined using DRG tariffs, which represent the reimbursement levels by the INHS to healthcare providers. Healthcare costs related to specialist visits, and diagnostic services were defined according to the tariffs of each region (called Nomenclatore tariffario regionale).

### 2.5. Statistical Analysis

All analyses were descriptive. Categorical variables have been reported as numbers and percentages, continuous variables as mean with SD. Patients with values exceeding the mean value three times the SD were excluded from the cost analysis. Following the “Opinion 05/2014 on Anonymization Techniques” drafted by the “European Commission Article 29 Working Party”, the analyses involving ≤ 3 patients were not reported (NR) for data privacy, as they were potentially traceable to single individuals. All analyses have been performed using STATA SE version 12.0.

## 3. Results

From a sample population of around 2.7 million health-assisted subjects, 105,948 users of ophthalmic drops were identified, and among them 18,161 patients had evidence of glaucoma based on the criteria applied and were therefore included (Figure 1). 

Characteristics were reported in Table 1: 44% of patients was male and mean age was 67 years. The most populated age ranges were those 65−74 years (28.1%), 75−84 years (23.6%) and 55−64 years (20.4%). Mean Charlson Index was 0.9, with around 22% of patients showing a score ≥2 indicating a moderate-severe comorbid profile. Hypertension was the comorbidity most frequently detected (60.2%) followed by dyslipidemia (29.7%) and diabetes (17%). Regarding eye-related diseases, cataract was observed in 8.9% of patients while 0.7% was blind. At index date, 30.3% of patients received prostaglandin analogues, 30% had fixed combination, mostly timolol-based, while 25.7% received beta blocking agents, 10.8% carbonic anhydrase inhibitors and 3% sympatico mimetics (Figure 2). 

During all the period analyzed, 11.5% of patients had a trabeculectomy, 2% a trabeculoplasty. Patients that underwent trabeculoplasty were older compared to those treated with drops only and showed a higher level of comorbidity profile (Table 1).

Of all the patients included (*N* = 18,161), by considering all available period, 70% (*N* = 12,754) had a second line of therapy and 57% (*N* = 10,394) a third line. Lines of therapy were mainly represented by ophthalmic drugs, and therapeutic sequences are reported in Table 2. As first line, 96.3% patients had ophthalmic drops, while only a small proportion of patients reported trabeculectomy (3.5%) or trabeculoplasty (0.4%). The majority of patients (66%) with ophthalmic drops as first line switched to another ophthalmic therapy, while 2.8% had a trabeculectomy procedure and 0.4% a trabeculoplasty. All patients with trabeculectomy or trabeculoplasty as first line had ophthalmic drops as second line, while as third line a second procedure was found, respectively, in 11.6% and 13% of patients (Table 2). 

Regarding drug utilization, during first year of follow-up, 58.3% were adherent to ophthalmic drops, 25.6% partially adherent and 16.1% poorly adherent (Table 3). Persistence to ophthalmic medication interested 78.1% of patients while the remaining 21.9% interrupted the therapy. Around 42% of patients switched the index ophthalmic drugs during the first year of follow-up.

The analysis on mean annual resource consumption and costs during first year of follow-up revealed a mean annual number of 17 prescriptions, 6.4 outpatient specialist services and a mean of 0.3 all-cause hospitalization. The mean total annual direct cost per patients was €1,725, related mostly to all-cause drug expenditure (€800) followed by all-cause hospitalizations (€567) and outpatient services (€359) (Figure 3).

## 4. Discussion

This analysis on real-world data provided insights into characteristics of glaucoma patients, their therapeutic paths, and health care resource use and related direct costs for INHS in Italian settings of clinical practice. Among 2.4 million health-assisted individuals, almost 18,000 glaucoma patients under ophthalmic drops were included in the analysis, with a prevalence of 0.67%. In Europe glaucoma prevalence is almost 2.5% [6], and in Italy 550,000 individuals are estimated to have received a diagnosis for glaucoma [7], and prevalence rates of 2.51% of Primary Open Angle Glaucoma, 0.97% of Primary Closed Angle Glaucoma and 0.29% of secondary glaucoma were estimated [19]. The discrepancy between our data and published reports is feasibly attributable to the fact that in the present analysis glaucoma patients were identified by treatment prescription and not by a direct diagnosis. 

The analysis of patient’ characteristics revealed a mean age of 66 years and almost 60% being female; these data were in line with other real-world studies reporting the same mean age and a slight female predominance [6]. The comorbidity profile of these patients showed a higher frequency of hypertension and dyslipidemia in almost 20−30% of patients; data from an observational Italian study reported the most frequent comorbidities (self-reported) were systemic hypertension (53.2%) and hyperlipidemia (26.2%), similar to our findings [9]. All these comorbidities indicate a polypharmacy tendency for these patients, suggesting paying attention to avoid drug–drug interaction in patients prescribed multiple drugs and that an individualized management should be considered that integrates anti-glaucoma agents into the overall treatment plan [20]. In the present analysis all glaucoma patients under ophthalmic drops were included, being by definition in first-line treatment. Most of the patients (86.7%) were under ophthalmic drops as monotherapy, as per guidelines [10]. 

It has been extensively reported that adherence to glaucoma medication could be a challenging problem [12]. Adherence to ophthalmic medication is poor, and multiple factors have been identified, including more recurrent and complex dosing, as well as patient-centered factors, such as poor disease or health consciousness, and a passive learning style [21]. Medication adherence plays an essential role alongside several factors such as clinical benefit, economic burden and quality of life of a patient [18,22,23]. In our analysis we have found that 58.3% were adherent, 25.6 % partially adherent and 16.1% poorly adherent. The latter value is within the rates of nonadherence with glaucoma medications found in the literature, that span from 16% to 30% [24]. It should be underlined that our analysis was limited to one year of observation but given that glaucoma is a chronic condition requiring a life-long treatment, studies with longer follow-up have shown that therapy adherence tends to further decrease over the years [23]. Similarly, persistence also ranged from 69% to 84%, according to European studies [24,25]. A proper drug utilization, namely optimization of adherence and persistence to treatment, may provide a decrease in the healthcare burden of patients. 

The analysis of healthcare resource consumption and cost showed that medication expenditures were found to be the main driving force, accounting for 46% of total costs. In other European counties, treatment costs for patients with glaucoma has been reported to range between 42%−56% of total direct cost for patients in all stages of glaucoma [23]. Moreover, it has been reported that the economic burden of glaucoma increases with disease severity. An analysis performed in Europe showed an increase of around €86 on total cost for each progression in glaucoma stage, from €455 (stage 0) to €969 (stage 4) per person year [26].

The present analysis has some limitations related to its observational and retrospective design and to the data source. Indeed, administrative databases are primarily intended for administrative purpose, even if their utilization for healthcare research is increasing over the years. Some limitations are related to the lack of clinical data within the database therefore, it was not possible to retrieve information on the status of glaucoma, level of severity, nor type of glaucoma. Furthermore, the identification of patients was made by presence of ophthalmic drugs; therefore, untreated patients were not captured. The comorbidities were observed during all data availability periods before inclusion; therefore, variations and incomplete capture of these variables could have been present among patients. Drug utilization is based on drug dispenses; therefore, reasons behind choice of therapy or switch are not collected. Minimally invasive glaucoma surgery (MIGS) was not identified as, to date, there is no code for reimbursement structure for MIGS available in Italy [7].

## 5. Conclusions

This real-world analysis depicted the characteristics, therapeutic path and economic burden of glaucoma patients under ophthalmic drops in Italy by means of administrative data. The vast majority of treated patients were under ophthalmic medication in monotherapy. Drug utilization analyses reveal poor adherence and persistence below 80%. Results were consistent with the literature, while the low prevalence reported could be explained by the methodology applied, since the analysis focused on glaucoma patients in treatment. Patients’ management was associated with healthcare resource consumption and costs mostly related to drug prescriptions. Although this result could depend on the fact all patents were treated, this trend adds to the growing body of knowledge that treatments are a major driving force for glaucoma patients. Overall, these real-life data advise that strategies to optimize glaucoma management should be focused on ensuring a proper drug utilization; efforts to increase the adherence and persistence to ophthalmic medication has been widely reported to enhance the likelihood to get benefit from the therapy. Furthermore, we have shown a complex therapeutic pattern for these patients, that move towards multiple line of therapy, and, in addition, displayed a comorbid profile requiring a polytherapy regimen with risk of drug-drug interaction, suggesting an unmet therapeutic need that should be taken into account in the development of new treatments/techniques for glaucoma.

## Figures and Tables

**Figure 1 healthcare-11-00635-f001:**
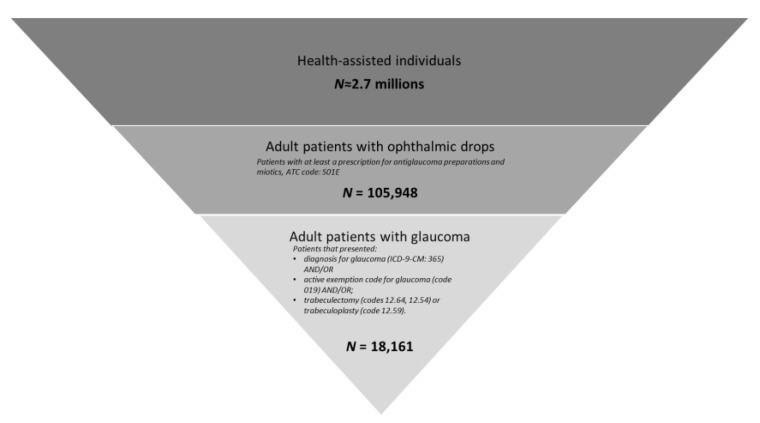
Flow-chart of the included patients based on the methodology applied.

**Figure 2 healthcare-11-00635-f002:**
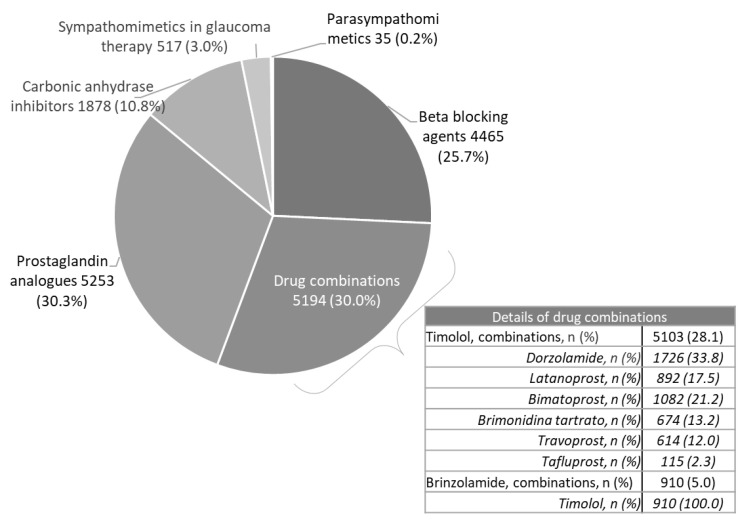
Class of ophthalmic drops treatments at the index date.

**Figure 3 healthcare-11-00635-f003:**
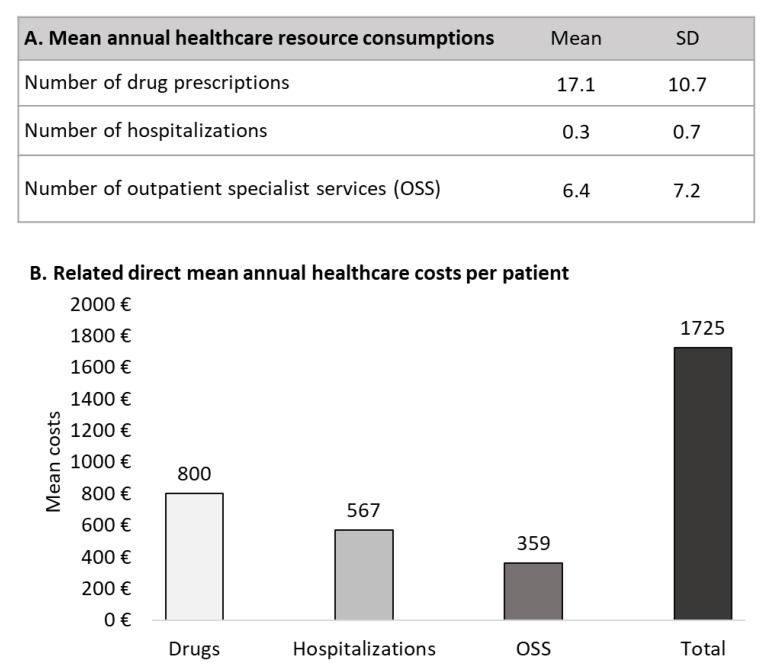
Mean annual healthcare resource consumptions (**A**) and related direct costs (**B**) over first year of follow-up in patients with glaucoma. The index-date was included in the cost analysis.

**Table 1 healthcare-11-00635-t001:** Characteristics of glaucoma patients.

	Total Glaucoma Patients (*N* = 18,161)	Patients with Drop Therapy Only (*N* = 15,749)	Patients with Trabeculoplasty (*N* = 354)	*p*-Valued
Male, n (%)	8047 (44.3)	7151 (45.4)	201 (56.8)	<0.001
Age, mean (SD)	66.6 (14.7)	68.6 (13.0)	70.5 (11.6)	0.006
Age groups				
18−24 years, *n* (%)	137 (0.8)	51 (0.3)	NR	
25−34 years, *n* (%)	508 (2.8)	145 (0.9)	NR	
35−44 years, *n* (%)	897 (4.9)	494 (3.1)	5 (1.4)	0.002
45−54 years, *n* (%)	1902 (10.5)	1661 (10.5)	24 (6.8)	
55−64 years, *n* (%)	3704 (20.4)	3363 (21.4)	62 (17.5)	
65−74 years, *n* (%)	5091 (28.1)	4538 (28.8)	114 (32.2)	
75−84 years, *n* (%)	4276 (23.6)	3923 (24.9)	117 (33.1)	
>85 years, *n* (%)	1646 (9.1)	1574 (10.0)	29 (8.2)	
Charlson index (mean, SD)	0.9 (1.0)	0.9 (1.0)	1.1 (1.1)	<0.001
Charlson index = 0, *n* (%)	7743 (42.6)	6744 (42.8)	115 (32.5)	<0.001
Charlson index = 1, *n* (%)	6481 (35.7)	5645 (35.8)	141 (39.8)	
Charlson index ≥ 2, *n* (%)	3937 (21.7)	3360 (21.3)	98 (27.7)	
Hypertension, *n* (%)	10,935 (60.2)	9861 (62.6)	250 (70.6)	0.002
Dyslipidemia, *n* (%)	5385 (29.7)	4868 (30.9)	127 (35.9)	0.046
Diabetes, *n* (%)	3091 (17.0)	2733 (17.4)	115 (32.5)	<0.001
Cataract, *n* (%)	1609 (8.9)	1398 (8.9)	53 (15.0)	<0.001
Blindess, *n* (%)	126 (0.7)	99 (0.6)	NR	-
Retinal/choroid disorders, *n* (%)	301 (1.7)	222 (1.4)	16 (4.5)	<0.001
Diabetic retinopathy, *n* (%)	72 (0.4)	57 (0.4)	NR	-
Wet age-related macular degeneration, *n* (%)	22 (0.1)	16 (0.1)	NR	-
Retinal vein occlusion, *n* (%)	24 (0.1)	15 (0.1)	NR	-
Parkinson’s disease, *n* (%)	98 (0.5)	96 (0.6)	NR	-
Alzheimer’s disease, *n* (%)	24 (0.1)	23 (0.1)	NR	-
Rheumatoid arthritis, *n* (%)	168 (0.9)	158 (1.0)	NR	-

Note. NR: not reported for data privacy (<4 patients).

**Table 2 healthcare-11-00635-t002:** Sequencies of therapeutic lines in patients with glaucoma.

First Line	Second Line	*N* (%)	Third Line	*N* (%)
Ophthalmic drops (N = 17,456) - N = 13,466: single agent (SA) - N = 3990: combination (COMB)	/	5407 (31.0)		
	Other ophthalmic drops 1st L SA -> 2nd L SA 1st L SA -> 2nd L COMB 1st L COMB -> 2nd L SA 1st L COMB -> 2nd L COMB	11,490 (65.8) −5644 3231 −1951 −664	/	1627 (14.2)
			Other ophthalmic drops - 2nd L SA -> 3rd L SA - 2nd L SA -> 3rd L COMB - 2nd L COMB -> 3rd L SA - 2ndL COMB -> 3rd L COMB	9733 (84.7) - 4609 - 1902 - 2623 - 599
			Trabeculectomy	102 (0.9)
			Trabeculoplasty	28 (0.2)
	Trabeculectomy	481 (2.8)	/	276 (57.4)
			Other ophthalmic drops	205 (42.6) (164 SA, 41 COMB)
	Trabeculoplasty	78 (0.4)	/	29 (37.2)
			Other ophthalmic drops	48 (61.5) (36 SA, 12 COMB)
			Trabeculectomy	NI
Trabeculectomy (*N* = 628)	Ophthalmic drops	628 (100) (518 SA, 110 COMB)	/	389 (61.9)
			Other ophthalmic drops 2nd L SA -> 3rd L SA 2nd L SA -> 3rd L COMB 2nd L COMB -> 3rd L SA 2ndL COMB -> 3rd L COMB	166 (26.4) - 80 - 47 - 29 - 10
			Trabeculectomy	73 (11.6)
Trabeculoplasty (*N* = 77)	Other ophthalmic drops	77 (100) (71 SA, 6 COMB)	/	39 (50.6)
			Other ophthalmic drops 2nd L SA -> 3rd L SA 2nd L SA -> 3rd L COMB 2nd L COMB -> 3rd L SA	28 (36.4) - 20 - 4 - 4
			Trabeculoplasty	10 (13.0)

**Table 3 healthcare-11-00635-t003:** Drug utilization over first year of follow-up in patients with glaucoma.

	Patients with Glaucoma (*N* = 18,161)	
	*N*	%
**Adherence to treatment**		
PDC ≥ 80	10,581	58.3
40 ≤ PDC < 80	4651	25.6
PDC < 40	2928	16.1
**Persistence to treatment**	14,179	78.1
Discontinuation to treatment	10,685	58.8
Interruption	3982	21.9
Switch	7605	41.9

Note. PDC: Portion of Day Covered.

## Data Availability

All data used for the current study are available upon reasonable request to CliCon S.r.l. Società Benefit, which is the body entitled to data processing by the other participating entities.

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
