# Peer review of "Real-World Analysis on the Characteristics, Therapeutic Paths and Economic Burden for Patients Treated for Glaucoma in Italy"

_healthcare, 2023, doi:10.3390/healthcare11050635_

Round 1

Reviewer 1 Report

The paper is a comprehensive analysis of the healthcare expenditure associated with glaucoma in Italy. The study is well designed and clearly presented.

As a minor issue: please add a paragraph of possible ways to decrease the heathcare burden associated with glaucoma and avoid the risc of polymedication

Author Response

Reviewer 1

The paper is a comprehensive analysis of the healthcare expenditure associated with glaucoma in Italy. The study is well designed and clearly presented.

Reply: We thank the Reviewer for the positive feedback.

As a minor issue: please add a paragraph of possible ways to decrease the heathcare burden associated with glaucoma and avoid the risc of polymedication

Reply: we thank the Reviewer; we have added a couple of sentences accordingly.

Reviewer 2 Report

The authors have presented the analysis on Characteristics, Therapeutic Paths and Economic Burden for Patients Treated for Glaucoma in Italy. The manuscript needs the following changes to be more effective.

General

1. Plagiarism is high (29% using 4 words at a time on Turnitin), it must be corrected.

2. Title seems to be awkward, one of the suggestion could be using Real-world analysis............

Technical

1. Method of shortlisting the articles, notes, studies and selection of usable data with adopted constrains must be clearly shown in a diagram/block-diagram to generate a clear understanding of the adopted methodology

2. There are instances where assumptions have been made regarding the interpretation of available data, authors must defend their stance properly at the same point

3. Conclusion needs to be re-written under the constraints of assumptions made as many findings of study are quite obvious and the contrasting outcome facts must be highlighted properly  

Author Response

The authors have presented the analysis on Characteristics, Therapeutic Paths and Economic Burden for Patients Treated for Glaucoma in Italy. The manuscript needs the following changes to be more effective.

General
1. Plagiarism is high (29% using 4 words at a time on Turnitin), it must be corrected.

Reply: we have modified the text accordingly. Please note some sentences in the methodology section may result similar to other studies since these are standard statements for data privacy, or information stored in the administrative DB, or are referred to the codes used to identify the comorbidities and they cannot be much modified.

  1. Title seems to be awkward, one of the suggestion could be using Real-world analysis............

Reply: the title has been modified as suggested.

Technical

  1. Method of shortlisting the articles, notes, studies and selection of usable data with adopted constrains must be clearly shown in a diagram/block-diagram to generate a clear understanding of the adopted methodology

Reply: we thank the Reviewer for this comment. We want to clarify this is not a systematic review based on published literature; as reported in the paragraph 2.1, data are retrieved from administrative databases of a sample of local health units in Italy, and we have provided a brief description on what are the info collected within this database. We have reported the flowchart of the study with the codes used, hoping to improve the clarity of the methodology adopted.

  1. There are instances where assumptions have been made regarding the interpretation of available data, authors must defend their stance properly at the same point

Reply: we thank the Reviewer for this comment: we had already inserted a limitation paragraph to report all of the limitation of the analysis. We have revised this part and stressed more in the text where assumptions were made in the interpretation of available data.

  1. Conclusion needs to be re-written under the constraints of assumptions made as many findings of study are quite obvious and the contrasting outcome facts must be highlighted properly .

Reply: the conclusions were modified accordingly. Please note in the discussion section we have compared our results with the literature, and they were quite consistent. When the results were contrasting, we have provided an explanation. We have reported this part in the conclusions.

Reviewer 3 Report

This is a very useful presentation of the grouped approach to glaucoma by Ophthalmologists in Italy. It is written in Methods:'' The present retrospective observational analysis was based on data collected from 80 the administrative databases of a sample of Italian Local Health Units (LHUs), covering 81 around 2.7 million health-assisted subjects''. Could you be more precise? Which Units, regions and how many Ophthalmologists were involved?

Can you please make an appendix with the raw data for consultation?

It is good to include, as you did, a paragraph on the limitations of the database. To better understand this limitations, I would describe in a few lines how the system is constructed as in the discussion now there is only this sentence:''Region/LHUs administrative databases have progressively improved the quality of the collected data'' How? ''in spite of some missing information that might have reduced the numerosity of the population.'' which information? ''Some limitations are related to the lack of clinical data within the database therefore it was not possible to retrieve information on the status of glaucoma, level of severity, nor type of glaucoma''.

Author Response

This is a very useful presentation of the grouped approach to glaucoma by Ophthalmologists in Italy. It is written in Methods:'' The present retrospective observational analysis was based on data collected from 80 the administrative databases of a sample of Italian Local Health Units (LHUs), covering 81 around 2.7 million health-assisted subjects''. Could you be more precise? Which Units, regions and how many Ophthalmologists were involved?

Reply: we thank the Reviewer for the positive comment. Actually, the data were collected from the Local Health Units and are based on the administrative databases, that contain all data reimbursed by the INHS. That allows to collect evidence on the “real-world”, ie patients not necessarily referred to specialized centers. Therefore, there is no direct involvement of ophtalmologists. We have however provided the Regions the LHUs are located.

Can you please make an appendix with the raw data for consultation?

Reply: we thank the Reviewer, however we cannot share the raw data as per privacy and agreements with the Local health units, as reported in the data availability statement.

Round 2

Reviewer 2 Report

Satisfied with the incorporated changes. Plagiarism is still high (25%). Authors are advised to further work on same, especially on sources 1 and 2. Please check the attached file for your reference.

Author Response

We thank the Reviewer for this comment, we have reworded parts of the manuscript accordingly. Please kindly note, sources 2 detects as plagiarism the template of the word file, which of course we cannot change, same with the ATC/ICD-9 code that were detected as plagiarism.